# A Review and Perspective of eDNA Application to Eutrophication and HAB Control in Freshwater and Marine Ecosystems

**DOI:** 10.3390/microorganisms8030417

**Published:** 2020-03-16

**Authors:** Qi Liu, Yun Zhang, Han Wu, Fengwen Liu, Wei Peng, Xiaonan Zhang, Fengqin Chang, Ping Xie, Hucai Zhang

**Affiliations:** 1Institute for Ecological Research and Pollution Control of Plateau Lakes, School of Ecology and Environmental Science, Yunnan University, Kunming 650504, China; liuqi@ynu.edu.cn (Q.L.); zhangyun@ynu.edu.cn (Y.Z.); wuhan@ynu.edu.cn (H.W.); liufw@ynu.edu.cn (F.L.); pengw@ynu.edu.cn (W.P.); zhangxn@ynu.edu.cn (X.Z.); changfq@ynu.edu.cn (F.C.); xieping@ihb.ac.cn (P.X.); 2Donghu Experimental Station of Lake Ecosystems, State Key Laboratory of Freshwater Ecology and Biotechnology, Institute of Hydrobiology, CAS, Wuhan 430072, China

**Keywords:** eDNA, eutrophication, harmful algae blooms, freshwater ecosystem, marine ecosystem

## Abstract

Changing ecological communities in response to anthropogenic activities and climate change has become a worldwide problem. The eutrophication of waterbodies in freshwater and seawater caused by the effects of human activities and nutrient inputs could result in harmful algae blooms (HABs), decreases water quality, reductions in biodiversity and threats to human health. Rapid and accurate monitoring and assessment of aquatic ecosystems are imperative. Environmental DNA (eDNA) analysis using high-throughput sequencing has been demonstrated to be an effective and sensitive assay for detecting and monitoring single or multiple species in different samples. In this study, we review the potential applications of eDNA approaches in controlling and mitigating eutrophication and HABs in freshwater and marine ecosystems. We use recent studies to highlight how eDNA methods have been shown to be a useful tool for providing comprehensive data in studies of eutrophic freshwater and marine environments. We also provide perspectives on using eDNA techniques to reveal molecular mechanisms in biological processes and mitigate eutrophication and HABs in aquatic ecosystems. Finally, we discuss the feasible applications of eDNA for monitoring biodiversity, surveying species communities and providing instructions for the conservation and management of the environment by integration with traditional methods and other advanced techniques.

## 1. Introduction

The eutrophication of lakes and coastal oceans, which is caused by excess inputs of nutrients, has become a worldwide problem [1]. Nutrient inputs are produced by anthropogenic activities, including urban development and the agriculture, forestry, and fishery industries [2]. The main cause of lake eutrophication is increasing nitrogen (N) and phosphorus (P) inputs [3,4]. Reducing N and P concentrations can help to improve water quality and ecological status [5,6]. The negative effects of eutrophic waters might result in murky water, harmful algae blooms (HABs) caused by phytoplankton, anoxia or toxicity of waterbodies and reduced species diversity [7]. Cyanobacterial blooms, such as those caused by *Microcystis*, are a major sign of eutrophication in lakes and can destroy the food web in aquatic environments because of their toxicity and rapid proliferation [8]. It has been demonstrated that long-term exposure to microcystin could promote the development of cancer, such as liver cancers, and be harmful to human health [9]. 

There are many physical and chemical techniques to control cyanobacterial blooms and improve water quality, such as coagulation and treatment with copper sulfate [10,11]. However, these physical and chemical methods can be time-consuming, expensive, and damaging to the environment. Using chemical methods might induce heavy metal accumulation and secondary pollution of the aquatic environment because of the stability and residues of the chemical reagents [12]. Biological approaches have been recommended as a useful tool in cyanobacterial bloom control [11]. It has been widely demonstrated that algicidal microorganisms such as bacteria, fungi and viruses can inhibit cyanobacterial blooms effectively and with specificity [13,14,15].

Environmental DNA (eDNA) has been reported to be a useful tool for monitoring species distributions and identifying different organisms [16]. eDNA is composed of extracellular DNA released by organisms through discharges such as secretions, feces, gametes, sloughed cells, hair, or bodily remains in the environment [17,18]. Ogram et al. introduced eDNA in the field of microbiology in 1987 [19]. eDNA is a valuable tool, and analysis of eDNA is increasingly being used to identify sensitive and invasive species or pathogens, detect low abundance or endangered species and estimate biodiversity [20,21] (Appendix A). Furthermore, previous studies indicated that eDNA analysis could be more convenient, efficient and comprehensive than conventional methods for analyzing different taxa across spatial and temporal scales [22].

In aquatic ecosystems, traditional methods, such as physical identification, are inadequate for reflecting the real population structure and detecting invasive species early [23], because the organisms in aquatic environments could be hidden underwater and difficult to find by conventional techniques. eDNA has a better applicability to a wider range of water bodies and other animals or plants [24]. Therefore, eDNA analysis could be a sensitive and effective assay for detecting species that are undetectable by traditional survey approaches [25,26]. In addition, eDNA analysis contributes positively to the detection of certain groups from complex samples in aquatic ecosystems. For example, Rudko et al. developed four species-specific detection assays in water samples from lakes in Northern Michigan [27]. Many studies have reported that eDNA has been used in the investigation of freshwater and marine ecosystems. As suggested by Banerji et al., DNA metabarcoding supplements conventional biological methods to obtain more comprehensive profiles of freshwater plankton community structure and distribution [28]. The rapid monitoring of changes in biodiversity using eDNA holds great promise for monitoring the changes expected as a result of climate change, especially in large lake ecosystems [29]. On the other hand, eDNA analysis is also used to explore and survey microorganisms, fishes or other species in seawater samples [30,31,32].

Furthermore, for environmental management and ecological research, eDNA is able to integrate traditional and advanced techniques for surveying aquatic ecosystems and biological interactions with different environmental factors [33,34]. In this review, we describe the potential applications of eDNA methods for eutrophication and HAB control in freshwater and marine ecosystems.

## 2. eDNA Methods for Eutrophication and HAB Control

It is widely cognized that eDNA metabarcoding could contribute to the development of biological techniques for use against eutrophication and HABs, such as cyanobacterial blooms. We obtained the total number of publications in the Web of Science database from 1975 to 2020 (2nd March) with the terms “eDNA, eutrophication and HAB/s” in the title, abstract or keywords based on bibliometric methods (Appendix A) [35]. There were 469 publications in this period (Figure 1). This analysis indicated that before 2013, the publication output remained at a low level, which meant that the eDNA approach did not receive enough attention to survey eutrophication and HABs. After 2013, however, this topic was increasingly researched. The key reason for this increase might be the global rapid economic development, which may have resulted in increasing eutrophication in waterbodies [36] and the rapid development of DNA sequencing technology, which provided effective and accurate methods to conduct DNA research at the molecular level. Rapid economic development and human activities are the major source of nutrients, such as increasing nutrient loading, which can cause the problem of eutrophication. Increasing eutrophication is expected to amplify the severity of the ecological problems. Advanced technologies are employed to deal with the eutrophication and HAB control. In the field of nucleic acid research, there are a lot of technological and scientific breakthroughs observed within the last few years. The inputs from large numbers of researchers around the world who have invested many time and resources to develop technologies that can innovate the DNA sequencing strategies [37]. For example, the generations of DNA sequencing have moved on from the first generation to the third generation. These changes in DNA sequencing technology has altogether extended the potential outcomes of employing eDNA and is expected to supplement the traditional methods. Undoubtedly, eDNA research is gaining momentum swiftly because of the developments in sequencing techniques [38]. The new strategy of integrating eDNA techniques into eutrophication and HAB control has attracted more and more attention worldwide.

Seven document types were identified among the documents collected from the database (Figure 2 and Appendix A). Articles were the main documents type, comprising 87.80% of the total publications, followed by reviews, accounting for 8.71% of the total. This finding suggested that using eDNA to research eutrophication and HABs was at an early stage and would later receive more attention. In addition, a total of 41 countries were covered by the publications obtained from the Web of Science database. As shown in Figure 3 and Appendix A, most of the published documents were from the United States, accounting for 30.49% of the total, followed by Germany, the United Kingdom, Japan and Australia, accounting for 9.17%, 8.96%, 6.18%, and 4.69% of the total, respectively. This finding indicated that a relatively greater number of countries have conducted research exploring the application of eDNA (which is an environmentally friendly method based on advanced technology) in the field of eutrophication and HAB control. Increasing eDNA studies for mitigating and controlling eutrophication and HABs could improve water quality, helping to protect the environment and manage aquatic ecosystems.

### 2.1. Freshwater Ecosystems

#### 2.1.1. eDNA and Eutrophication in Freshwater

Many studies have focused on monitoring and managing freshwater environments, including lakes, ponds and rivers [40,41,42]. eDNA could be used to identify environmental stressors and evaluate the trends of lake ecosystems by accurately monitoring changes in the microorganism and plankton community structure in lakes [43]. It has been reported that eDNA data might be helpful for predicting pollution status and uncovering the main factors affecting ecological networks. Li et al. determined species distributions and main stressors of the community structure, such as total nitrogen (TN) and total phosphorus (TP), through eDNA metabarcoding and principal component analysis (PCA) in the Yangtze River Delta (YRD) [42]. These results could be used to predict the pollution status based on the operational taxonomic units (OTUs) and multivariate linear regression models (MLRs).

Furthermore, the exchange of nutrients between waterbodies and sediments is important for altering the trophic level of water [44]. The biological transportation and metabolism of microorganisms in sediments could affect the cycling of N and P [45,46]. Therefore, bacteria in the sediment also play a critical role in aquatic ecosystems. High-throughput DNA sequencing was used to analyze the sediment bacterial community in Lake Taihu [47]. The data showed that biodiversity is positively correlated with NH_4_^+^-N in the water and negatively correlated with NO_X_-N in the sediment. On the other hand, land-use intensification threatens freshwater ecosystems because of the chemical contaminants discharged by different land-use types [48]. Many chemical contaminants have been reported to result in a decrease in eukaryotic microbial diversity and disturbance of the cycling of geochemical elements in water [49]. Xie et al. indicated that in situ sampling of eukaryotic communities could be used to reveal chemical pollution from different land-use types, such as agricultural regions and industrial regions, in sediments from Nanfei River in Anhui Province, China [50]. 

#### 2.1.2. eDNA and HABs in Freshwater

In eutrophicated water, cyanobacteria are dominant species that can form cyanobacteria blooms with rapid cell proliferation [51,52]. The global temperature rise and eutrophication increase might induce toxic blooms, such as cyanobacterial blooms [53]. The cyanobacterial blooms can have a negative influence on the environment and aquatic biodiversity by depleting oxygen and limiting the nutrient cycling and light [54]. Furthermore, the toxins produced by cyanobacteria have been reported to be a risk to animals, plants and human beings [55]. Bell et al. detected changes in the eukaryotic and bacterial communities and their relationships with the environmental conditions in Lake Velence [56]. There is a positive correlation between eukaryotic diversity, such as green algae and diatoms, and dissolved organic carbon (DOC) or dissolved inorganic carbon (DIC). Moreover, it has been shown that the *Microcystis* abundance has a negative correlation with bacterial (e.g., *Actinobacteria*) diversity. In Erhai Lake, Song et al. identified 473 *Microcystis* genotypes by seasonal monitoring using internal transcribed spacer (ITS) genes and amplification primers [57]. The data indicated that the early stage of eutrophication in Erhai Lake might result in high diversity of *Microcystis* due to the negative relationship between the diversity of ITS genotypes and TN/TP.

There are a great number of microbes such as bacteria, viruses and fungi, in freshwater ecosystems that play a critical role in controlling and mitigating HABs [58]. Heterotrophic bacteria have also been described for use in assessing and controlling the negative impacts of cyanobacteria. Berg et al. isolated 460 bacterial strains from lake, river, and drinking water samples [59]. They characterized the taxonomic properties of these heterotrophic bacteria by DNA sequencing based on 16S rRNA. The results of the study demonstrated the occurrence of many growth-promoting cyanobacterial strains, such as *Flavobacterium*, and growth-inhibiting cyanobacterial strains, such as *Pedobacter*. They also detected other strains that are able to degrade toxins or organic compounds such as *Sphingomonas*. The secondary metabolites produced by cyanobacteria might affect water quality and be harmful for plants, animals and humans [60,61]. High-throughput DNA sequencing can be used to discover new bacterial taxa which can degrade toxins or organic compounds and provide instructions to manage freshwater ecosystems.

eDNA is one of the most useful methods for detecting different species and exploring the relationship between biodiversity and environmental stressors, but few studies have investigated using eDNA to uncover algicidal microorganisms in freshwater environments. The major published documents have focused on isolated and cultured bacteria or fungi from freshwater environments that can inhibit HABs directly or indirectly. For example, growth-inhibiting bacteria (GIBs), which limit the growth of *Microcystis aeruginosa*, were isolated from the biofilm surface on different aquatic plants, such as *Trapa japonica* in Japan [14]. In Lake Biwa, Imai et al. discovered bacteria that caused mortality in *Microcystis* and GIBs from the biofilm surface of aquatic plants (*Egeria densa* and *Ceratophyllum demersum*) [55]. The data showed that *Agrobacterium vitis* which is the most powerful bacterial strain among the 37 isolated strains could inhibit the growth of *M. aeruginosa* cells significantly. In addition, Yu et al. identified that a *Streptomyces sp.* strain named HG-16 was able to inhibit harmful algae with high algicidal activity [62]. However, eDNA can monitor and detect more species than traditional clone isolation techniques. It is suggested that more than 90% of microorganisms from the environment cannot be isolated and are uncultivable in the laboratory [63]. Conventional methods could not detect most algicidal microorganisms and might create a bottleneck for the termination of HABs. Therefore, increasing attention should be paid to detecting and analyzing algicidal microorganisms by using the eDNA method.

### 2.2. Marine Ecosystems

#### 2.2.1. eDNA and Eutrophication in Seawater

The marine environment is important for ecosystem services such as food, medicine, and the regulation of the climate [64]. However, marine ecosystems suffer from multiple stressful factors, such as chemical pollution, eutrophication and harmful algae [65]. The effects of humans are among the driving factors for marine eutrophication and global climate change [66]. The eutrophication of marine waters could cause a reduction in water quality and have negative impacts on marine ecosystems, the economy, and human health [67]. There is an urgent need to monitor and assess the biodiversity of marine environments and decrease the eutrophic state of seawater. 

Effective aquatic management and monitoring depend on obtaining comprehensive information about the diversity and distribution of organisms. eDNA could help to develop monitoring and assessment programs from seawater samples [68]. Coastal ecosystems are threatened by the effects of anthropogenic activities such as marine transport [69]. In addition, it is difficult to identify ecological features due to the nonlinearity and complexity of coastal ecosystems [22]. However, Jo et al. used an eDNA approach to investigate the general ecological characteristics of aquatic biodiversity in the coastal ecosystem of Gwangyang Bay [22]. These results suggest that eDNA can be recommended as a useful tool for detecting spatial differences in species communities in coastal areas. As suggested by Cowart et al., eDNA metagenomic sequencing could evaluate biodiversity and ecosystems to more comprehensively supplement traditional approaches [70]. This study suggested that eDNA metagenomic sequencing could evaluate biodiversity and ecosystems to more comprehensively supplement traditional approaches. Furthermore, eDNA sequencing could assist in revealing the interactions between anthropogenic activity and environmental changes, especially in the nearshore region [71].

#### 2.2.2. eDNA and HABs in Seawater

HABs such as red tide can negatively influence marine environments and human health [72]. Different studies have reported that the growth of microalgae might be inhibited by certain macroalgae. For instance, the macroalgae *Enteromorpha clathrata* were demonstrated to significantly inhibit the growth of the red tide microalgae *Skeletonema costatum* [73]. It has been shown that three macroalgae *Ulva linza*, *Corallina pilulifera,* and *Sargassum thunbergii*, have growth-inhibiting effects on the red tide microalgae *Prorocentrum donghaiense* [74]. 

Microorganisms in the ocean are essential for global biomass and energy cycling. It has already been shown that the eDNA method could be applied to detect microbial communities from seawater samples [75]. The microbes that exist in freshwater ecosystems, are similar to those that exist in marine environments, which might enable the mitigation or termination of HABs. Many studies have demonstrated that a very large number of algicidal bacteria could influence algae blooms. In the coastal environment of southwestern Japan, 65 strains of algicidal bacteria and GIBs were identified by 16S rRNA sequencing [76]. It has been shown that the biofilm surface of seagrasses and seaweeds might provide an environment that harbors algicidal bacteria and GIBs [55,77]. Onishi et al. found that the GIBs isolated from the seagrass *Zostera marina* strongly inhibit the growth of the dinoflagellate *Alexandrium tamarense* [78]. Furthermore, algicidal bacteria and GIBs that have been shown to inhibit the red tide raphidophyte *Heterosigma akashiwo* and the dinoflagellate *Alexandrium tamarense* were also uncovered from seagrass and seaweed beds in the Puget Sound, USA [79].

The biological mechanisms of the inhibition of HABs by microorganisms or macroalgae requires further exploration through eDNA analyses. However, in marine ecosystems, the investigation of eDNA is considered to be more challenging than that in freshwater environments because of the large volume of water and certain abiotic factors, such as salinity and currents [80]. In the ocean, the concentrations of eDNA have been reported to be negatively associated with depth and increasing distance to the coast [81]. It has been documented that the concentrations of eDNA range from <1 µg/L in oligotrophic oceans [82] to 44 µg/L in subtropical estuaries [83]. eDNA has been proven to be a powerful and efficient tool for detecting large numbers of species and monitoring marine ecosystems. Therefore, the control and mitigation of HABs in the ocean are expected to benefit from the rapid rise in the eDNA technique.

## 3. Perspectives on eDNA

In recent years, eDNA analysis has drawn much attention as an effective and sensitive approach for researching biodiversity [84,85], which can help to analyze the changes of species in different eutrophication levels. The breakthroughs by and substantial development of high-throughput DNA sequencing have made it possible to use this method to rapidly detect organisms’ community structure [86]. In addition, eDNA metabarcoding has been proven to be useful for monitoring species communities both qualitatively and quantitatively [24]. The detection of species diversity, community structures and species change patterns are often of considerable significance in assessing the effectiveness of eutrophication and HAB control. For example, the biomass and species compositions of phytoplankton are important factors and indicators of lake water quality [87]. The analysis of species compositions can help discover ecological indicators to provide instructions to mitigate eutrophication and HABs in aquatic environments. The novel method of using eDNA as a molecular marker for species identification and biodiversity assessment could produce comprehensive and valuable information for the management and protection of ecosystems [88]. 

Traditional biological monitoring and surveys have most often relied on the morphological identification of taxa. However, there are some limitations to the morphological approaches for identifying aquatic biodiversity. For instance, they require a large amount of time to carefully prepare and preserve specimens. A large number of species or samples cannot be collected and observed directly. It is difficult to distinguish many closely related organisms visually because of the lack of diagnostic morphological characters [89]. Molecular genetic assessment based on DNA sequences could help researchers to obtain more information quickly and economically. DNA metabarcoding was used to investigate the spatial and temporal dynamics of planktonic organisms, such as fungi and green algae [28].

Using a single method, it is difficult to continue monitoring and making the assessments used to collect all the information on the samples from lakes or oceans. It has been suggested that the combination of eDNA and conventional methods is beneficial for monitoring waterbodies [90] because eDNA analysis can produce accurate information based on robust data related to population genetic parameters and relatively widespread taxa. Traditional approaches can provide information on morphological traits and population characteristics such as life stage, size and growth curves [24]. Therefore, it is necessary to integrate conventional methods with the eDNA approach to produce comprehensive and long-term data to help evaluate and manage ecosystems and policy decisions.

Furthermore, there are 2 cold topics and 4 hot topics according to the statistical data from the Web of Science publications (Figure 4 and Appendix A). The 2 cold topics, gene library and PCR assays, showed a decreasing trend in topic popularity from 1975 to 2020 (2nd March) (Figure 4a). This might be because the DNA library of a very large number of species and PCR technologies were not able to uncover most gene sequences when DNA sequencing first began. Therefore, many studies have focused on improving PCR methods and constructing gene libraries. It contributes to provide comprehensive data for further eDNA study. However, DNA sequencing technologies have been extensively developed in recent years. This might have led to the decreasing trend of these 2 cold topics. The 4 hot topics (species identification, biodiversity, aquatic species analysis and ecological communities) showed increasing trends in topic popularity from 1975 to 2020 (2nd March). This was especially true of species detection (Figure 4b). This result indicated that eDNA could be beneficial for detecting multiple species, monitoring the environment, analyzing biodiversity and evaluating ecosystems. Moreover, technologies in eDNA research have expanded to be able to analyze whole communities from a single sample [91]. The biodiversity assessment of whole communities needs to be applied to monitor the aquatic environments in different eutrophication levels and explore the relationship between algicidal microorganisms’ community and HABs.

Although the use of eDNA related to eutrophication and HABs has been studied in many countries in recent decades (Figure 3 and Appendix A), the current eDNA research rarely focuses on the application of eDNA for mitigating eutrophication and HABs based on topical data. Few investigations have reported the combination of eDNA and physical or chemical methods to reduce or inhibit eutrophication and HABs in aquatic ecosystems. Previous studies have documented the development and improvement of strategies for controlling eutrophication and HABs, such as physical, chemical, and biological strategies. The physical and chemical methods have been proven to be potentially harmful for aquatic ecosystems due to their high cost and the secondary pollution they cause [92]. It has been suggested that using biological approaches is effective and environmentally friendly [93]. To control eutrophication and HABs, N and P inputs must be reduced in eutrophicated aquatic systems [94]. Microorganisms are responsible for biogeochemical cycles in freshwater and marine environments. Therefore, eDNA could be used to find microorganisms that can efficiently utilize and transform nutrients, such as N and P, in waterbodies to decrease the concentration of these elements.

Moreover, algicidal microorganisms are recommended as a useful tool to reduce and mitigate HABs. It is also important to identify microbes that possess growth-inhibiting activity against harmful algae. It is widely believed that there are a great number of microorganisms with high algicidal activity in freshwater and marine environments that could be used to control HABs [14,62]. Furthermore, *Brevibacillus laterosporus* which was isolated from soil samples was reported to have algicidal capabilities [95]. Therefore, anthropogenic activities and environmental changes in lake basins and coastal areas can also affect water quality changes, which, in turn, affect water eutrophication levels and HABs.

Few studies have shown the mechanisms by which algicidal microorganisms terminate or decrease algae blooms. For instance, GIBs inhibit the motility of the raphidophyte *Chattonella antiqua* in two ways [76]: they can change the cell morphology of *C. antiqua* from spindle-shaped to roundish-shaped and they can elongate the process of cell division in *C. antiqua.* However, the molecular mechanisms underlying algicidal microorganisms are not yet clear. eDNA analysis is a promising tool, as it might provide a large amount of information about DNA sequences and accurate transcriptional information about specific genes or protein functions, which can be combined with metagenomics and transcriptomics to predict the function of different genes and proteins. Then, transgenic or gene editing technologies, such as CRISPR, could be used to improve the lysing or growth-inhibiting activity of algicidal microorganisms and enhance the function of specific genes that play an important role in the algicidal process. On the other hand, long-term monitoring of species biodiversity and environmental factors (such as pH, N, and P) is beneficial for understanding the optimal conditions for microbes. Early warning systems can be created and the inhibition of HABs can be achieved by artificially inducing the proliferation of those microorganisms before the outbreak of HABs.

The DNA sequencing analysis is possible to be aimed at detection of species for which the genome or a portion thereof exists in databases [85]. Reference DNA databases may limit eDNA application for some species, because eDNA methodologies can only provide the information of the organisms that are discoverable from genetic material. However, rapid advances about meta-data are being made to fill the gaps to some extent. The meta-data research is supported by many databases and programs, such as Genbank, the BarCode of Life, the International Barcode of Life (iBOL) program, and the Earth BioGenome Project (EBP) [85]. The metagenomics can be used to analyze the species which have not been added into databases. According to the meta-data, the taxa of these species can be identified based on genomic similarity. The biological function of these species can be predicted based on the homology of functional genes.

Overall, the temporal trend in the eDNA publications indicated continuous growth, based on the number of published papers (Figure 1). The growing tendency showed an increasing awareness of the key scientific points of eDNA. This suggests that the number of documents related to eDNA applications for controlling eutrophication and HABs will continue to increase in the coming years. This could provide new directions and challenges for future research in aquatic ecosystems (Figure 5).

## 4. Conclusions

With the rapid development of DNA sequencing technologies, eDNA has been proven to be highly successful for monitoring biodiversity and surveying species communities. There are a great number of microorganisms, animals and plants in freshwater and marine ecosystems. It is difficult to identify complete species communities and their distributions by conventional techniques. However, screening and finding algicidal microorganisms and other species at the molecular level can be accomplished more accurately and faster based on the alignment of DNA sequencing results in a database than with traditional methods. Moreover, microbial taxa and communities are changing with the environment and climate, with influences such as temperature, pH and seasonal alternation. eDNA can detect and monitor changes rapidly and in fine detail and provide comprehensive information at the global scale.

Continuous monitoring and detection of changes in freshwater and marine ecosystems are important for the control and mitigation of eutrophication and HABs. The integration of eDNA methods and other techniques, such as physical and chemical methods, could produce exhaustive data. Long-term data could provide pioneering advice and instructions for the conservation and management of the environment, especially in aquatic ecosystems. The sustainable management of aquatic ecosystems would provide a wide range of habitat conditions that are beneficial for freshwater and marine biodiversity as well as water quality improvement. These benefits are very valuable for global biodiversity, the economy and human health.

## Figures and Tables

**Figure 1 microorganisms-08-00417-f001:**
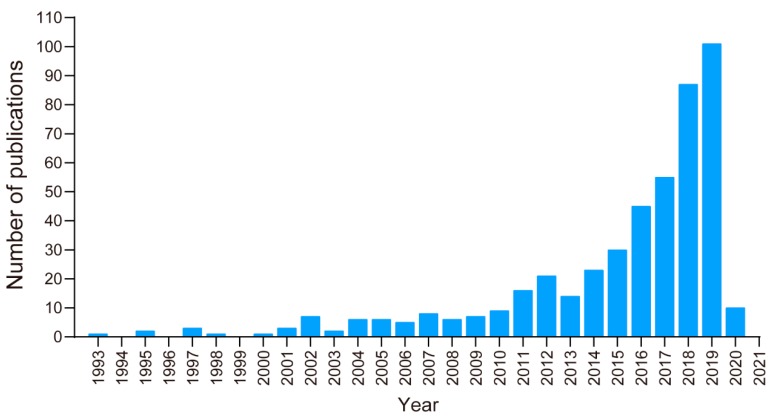
The number of journal publications with “environmental DNA (eDNA), eutrophication and harmful algae bloom/s (HAB/s)” in title, abstract and keywords during 1975–2020 (2nd March) in Web of Science. The data in the present work were sourced from the online database of the Science Citation Index Expanded (SCIE), Web of Science. This database is the most comprehensive and frequently used data source in bibliometrics [35,39]. The exact query was TS = ((“environmental DNA” OR “environment DNA” OR “eDNA”) AND (“eutrophication” OR “eutrophic” OR “nutrient*” OR “nitrogen” OR “phosphorus” OR “phosphate” OR “bloom” OR “blooms” OR “harmful algae bloom*” OR “HAB*”)), and the timespan was from 1975 to 2020 (2nd March).

**Figure 2 microorganisms-08-00417-f002:**
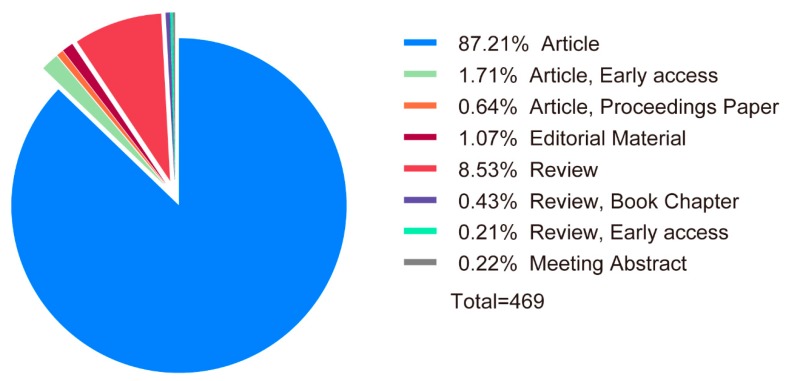
The percentages of seven types of documents.

**Figure 3 microorganisms-08-00417-f003:**
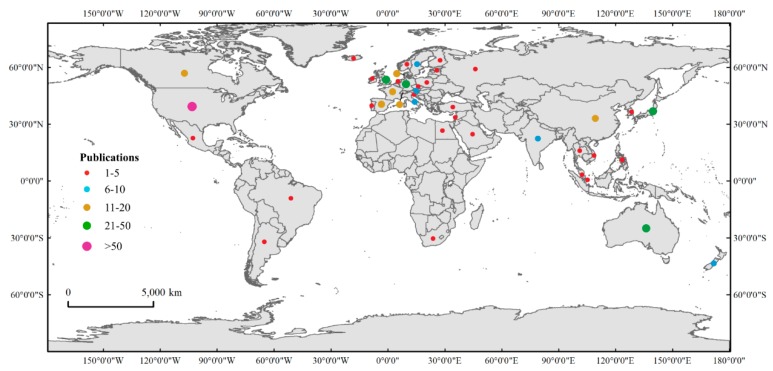
The worldwide distribution map of research on the use of eDNA with eutrophication and HABs from 1975 to 2020 (2nd March) based on publications from Web of Science.

**Figure 4 microorganisms-08-00417-f004:**
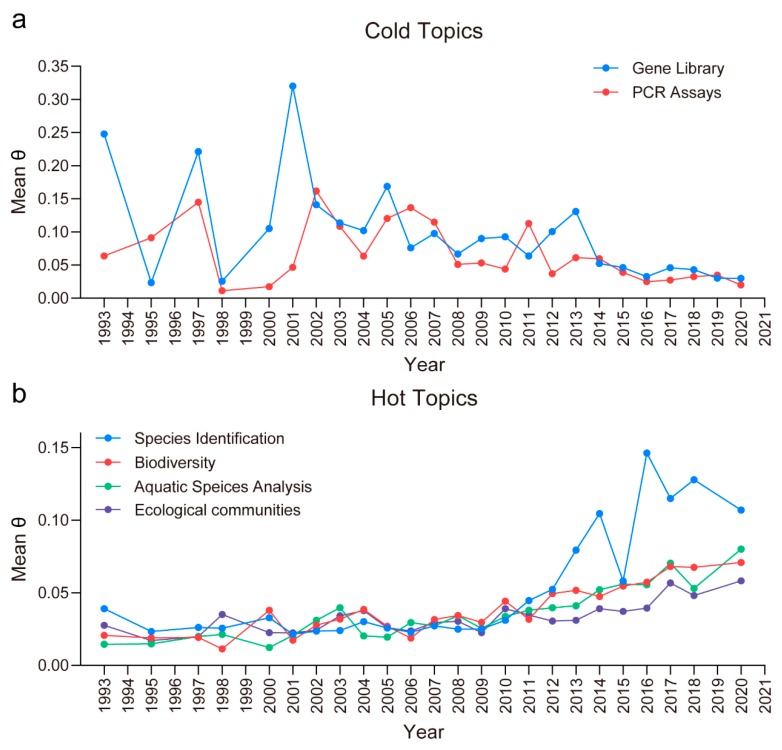
The cold (**a**) and hot (**b**) topics. The significant cold and hot topics sorted by the mean θ (the per-document probabilities for topics). The mean θ by year reflect the increasing and decreasing trends of scientific interests. In this study, we present a basic analysis based on a post hoc examination of the estimates of θ produced by the Latent Dirichlet Allocation (LDA) model [36]. To identify topics that were increasing and decreasing in popularity from 1975 to 2020 (2nd March), we conducted a linear trend analysis on θ of each topic by year. According to the mean θ and significance levels (*p* < 0.05), topics were identified as “cold” topics and “hot” topics.

**Figure 5 microorganisms-08-00417-f005:**
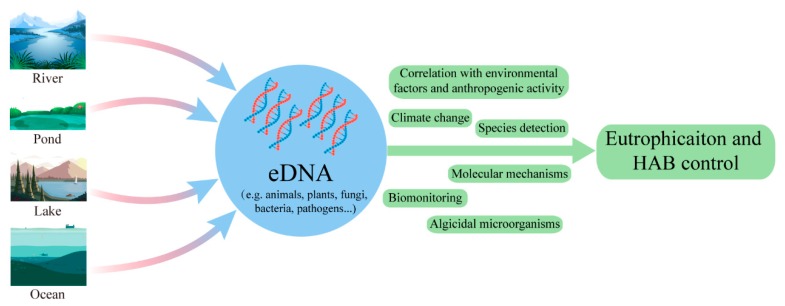
An overview of using eDNA as a tool for controlling and inhibiting eutrophication and HABs in aquatic ecosystems.

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
