# Peer review of "A Review and Perspective of eDNA Application to Eutrophication and HAB Control in Freshwater and Marine Ecosystems"

_microorganisms, 2020, doi:10.3390/microorganisms8030417_

Round 1

Reviewer 1 Report

Line 46:  not mental, rather metal

It is also unfocused on a single topic like HABs eDNA.  The background on fish and other eDNA is not needed. I also do not see value in geographic nature of eDNA studies or the increase in the number of studies.  

There is also no discussion of how DNA must exist in a database for a particular organism for it to be found with eDNA.  If a species has not been added.. extracted eDNA can do nothing to ID it.

Please update and search for newer literature on bloom genetic work being conducted

Author Response

Responses to Reviewer 1 Comments

Dear Reviewer 1,

We would like to express our sincere appreciation for your careful reading and invaluable comments to improve this manuscript. We have revised all issues raised by the reviewer. The line numbers in responses are based on the revised Microsoft word version. The revised parts are highlighted in yellow for Reviewer 1.

Point 1: Line 46: not mental, rather metal.

Response: Line 46: Thanks for your kind suggestion. We have corrected it in the revised manuscript.

Point 2: It is also unfocused on a single topic like HABs eDNA. The background on fish and other eDNA is not needed. I also do not see value in geographic nature of eDNA studies or the increase in the number of studies.

Response: Thanks for the suggestion. We have realized that the content related to fish and other eDNA may be confusing. The redundant content has been removed from the revised manuscript. In addition, we have updated and reanalyzed the data from the Web of Science database. The valuable data and eDNA studies which showed increasing trends have been shown in Figure 4.

Point 3: There is also no discussion of how DNA must exist in a database for a particular organism for it to be found with eDNA. If a species has not been added.. extracted eDNA can do nothing to ID it.

Response: Thanks for your kind advice. It is difficult to identify a species if the species has not been added in database. We have discussed the content of how DNA must exist in a database for a particular organism for it to be found with eDNA in the revised manuscript. We provide a practical way to ID the species based on the meta-data. Further discussions are presented as follows:

Line 671-680: The DNA sequencing analysis is possible to be aimed at detection of species for which the genome or a portion thereof exists in databases (Pikitch, 2018). Reference DNA databases may limit eDNA application for some species, because eDNA methodologies can only provide the information of the organisms that are discoverable from genetic material. However, rapid advances about meta-data are being made to fill the gaps to some extent. The meta-data research is supported by many databases and programs, such as Genbank, the BarCode of Life, the International Barcode of Life (iBOL) program, and the Earth BioGenome Project (EBP) (Pikitch, 2018). The metagenomics can be used to analyze the species which have not been added into databases. According to the meta-data, the taxa of these species can be identified based on genomic similarity. The biological function of these species can be predicted based on the homology of functional genes.

Point 4: Please update and search for newer literature on bloom genetic work being conducted.

Response: Thank you for the advice. We have researched and updated the data of newer literature on bloom genetic work in the revised manuscript, Figure 1, Figure 2, Figure 3, Figure 4 and Supplementary files.

Reviewer 2 Report

I think this review article is well written. I offer some comments about the overall organization and focus that the authors may wish to consider. Lines 89-92 seem to propose two hypotheses for trends the authors observed in publication rates. When I read this, I made note of it, expecting the authors to return to these points. I don't feel that the authors really did in any explicit way. I make this observation, in case it interests the authors.

There are a couple places where the authors use the phrase "universally agreed" (e.g. line 328). That is a pretty strong statement. Perhaps it would be better to say something like "widely agreed".

The title and abstract are really narrowly focused on the problems of eutrophication and HAB control. Section 3 - other applications of and perspectives on eDNA, seems to me to really stray away from that focus into more general territory on biodiversity sampling and so forth. There are lots of review articles on that topic. I think the authors have a more original contribution if they remain focused on the topics on these specific topics. I encourage the authors to consider trimming or elimination section 3.

Author Response

Responses to Reviewer 2 Comments

Dear Reviewer 2,

We would like to express our sincere appreciation for your careful reading and invaluable comments to improve this manuscript. We have revised all issues raised by the reviewer. The line numbers in responses are based on the revised Microsoft word version. The revised parts are highlighted in blue for Reviewer 2.

Point 1: I think this review article is well written. I offer some comments about the overall organization and focus that the authors may wish to consider. Lines 89-92 seem to propose two hypotheses for trends the authors observed in publication rates. When I read this, I made note of it, expecting the authors to return to these points. I don't feel that the authors really did in any explicit way. I make this observation, in case it interests the authors.

Response: Thanks for your kind suggestion. We have expanded the content of these two points to make them more clear in revised manuscript. Further discussions are presented as follows:

Line 90-146: Rapid economic development and human activities are the major source of nutrients, such as increasing nutrient loading, which can cause the problem of eutrophication. Increasing eutrophication is expected to amplify the severity of the ecological problems. Advanced technologies are employed to deal with the eutrophication and HAB control. In the field of nucleic acid research, there are a lot of technological and scientific breakthroughs observed within the last few years. The inputs from large numbers of researchers around the world who have invested many time and resources to develop technologies that can innovate the DNA sequencing strategies (Heather and Chain, 2016). For example, the generations of DNA sequencing have moved on from the first generation to the third generation. These changes in DNA sequencing technology has altogether extended the potential outcomes of employing eDNA and is expected to supplement the traditional methods. Undoubtedly, eDNA research is gaining momentum swiftly because of the developments in sequencing techniques (Garlapati et al., 2019). The new strategy of integrating eDNA techniques into eutrophication and HAB control has attracted more and more attention worldwide.

Point 2: There are a couple places where the authors use the phrase "universally agreed" (e.g. line 328). That is a pretty strong statement. Perhaps it would be better to say something like "widely agreed".

Response: Line 80 and 650: Thank you for the advice. We have corrected them in the revised manuscript.

Point 3: The title and abstract are really narrowly focused on the problems of eutrophication and HAB control. Section 3 - other applications of and perspectives on eDNA, seems to me to really stray away from that focus into more general territory on biodiversity sampling and so forth. There are lots of review articles on that topic. I think the authors have a more original contribution if they remain focused on the topics on these specific topics. I encourage the authors to consider trimming or elimination section 3.

Response: Thanks for the suggestion. We mainly focused on the areas of the problems of eutrophication and HAB control in eDNA research. The redundant content has been removed from the revised manuscript. In addition, our description may not be accurate according to reviewer’s suggestion. In this study, we describing eDNA analysis for researching biodiversity in section 3 is to highlight the possible applications of eDNA for eutrophication and HAB control. We considered that it can help to analyze the changes of species in different eutrophication levels and contribute to provide comprehensive data for further research. Therefore, we have modified the section 3 and made it streamlined in revised manuscript. Further discussions are presented as follows:

Line 389-391: In recent years, eDNA analysis has drawn much attention as an effective and sensitive approach for researching biodiversity (Kelly et al., 2014 , Pikitch, 2018), which can help to analyze the changes of species in different eutrophication levels.

Line 393-399: In addition, eDNA metabarcoding has been proven to be useful for monitoring species communities both qualitatively and quantitatively (Hanfling et al., 2016). The detection of species diversity, community structures and species change patterns are often of considerable significance in assessing the effectiveness of eutrophication and HAB control. For example, the biomass and species compositions of phytoplankton are important factors and indicators of lake water quality (Chen et al., 2017). The analysis of species compositions can help discover ecological indicators to provide instructions to mitigate eutrophication and HABs in aquatic environments.

Line 424-425: It contributes to provide comprehensive data for further eDNA research.

Line 431-620: Moreover, technologies in eDNA research have expanded to be able to analyze whole communities from a single sample (Ruppert et al., 2019). The biodiversity assessment of whole communities needs to be applied to monitor the aquatic environments in different eutrophication levels and explore the relationship between algicidal microorganisms’ community and HABs.